# On the effect of phylogenetic correlations in coevolution-based contact prediction in proteins

Edwin Rodriguez Horta[1,2], Martin Weigt[2]*

1 University of Havana, Physics Faculty, Department of Theoretical Physics, Group of Complex Systems and Statistical Physics, Havana, Cuba, 2 Sorbonne Université, CNRS, Institut de Biologie Paris-Seine, Laboratoire de Biologie Computationnelle et Quantitative – LCQB, Paris, France

* martin.weigt@sorbonne-universite.fr

**Data Availability Statement:** Data and code is available at https://github.com/ed-rodh/Null_models_I_and_II.

**Funding:** This work (MW) was funded by the EU H2020 Research and Innovation Programme MSCA-RISE-2016 under Grant Agreement No.

## Abstract

Coevolution-based contact prediction, either directly by coevolutionary couplings resulting from global statistical sequence models or using structural supervision and deep learning, has found widespread application in protein-structure prediction from sequence. However, one of the basic assumptions in global statistical modeling is that sequences form an at least approximately independent sample of an unknown probability distribution, which is to be learned from data. In the case of protein families, this assumption is obviously violated by phylogenetic relations between protein sequences. It has turned out to be notoriously difficult to take phylogenetic correlations into account in coevolutionary model learning. Here, we propose a complementary approach: we develop strategies to randomize or resample sequence data, such that conservation patterns and phylogenetic relations are preserved, while intrinsic (i.e. structure- or function-based) coevolutionary couplings are removed. A comparison between the results of Direct Coupling Analysis applied to real and to resampled data shows that the largest coevolutionary couplings, i.e. those used for contact prediction, are only weakly influenced by phylogeny. However, the phylogeny-induced spurious couplings in the resampled data are compatible in size with the first false-positive contact predictions from real data. Dissecting functional from phylogeny-induced couplings might therefore extend accurate contact predictions to the range of intermediate-size couplings.

## Author summary

Many homologous protein families contain thousands of highly diverged amino-acid sequences, which fold into close-to-identical three-dimensional structures and fulfill almost identical biological tasks. Global coevolutionary models, like those inferred by the Direct Coupling Analysis (DCA), assume that families can be considered as samples of some unknown statistical model, and that the parameters of these models represent evolutionary constraints acting on protein sequences. To learn these models from data, DCA and related approaches have to also assume that the distinct sequences in a protein family are close to independent, while in reality they are characterized by involved hierarchical

734439 InferNet. The funders had no role in study design, data collection and analysis, decision to publish, or preparation of the manuscript.

**Competing interests:** The authors have declared that no competing interests exist.

phylogenetic relationships. Here we propose Null models for sequence alignments, which maintain patterns of amino-acid conservation and phylogeny contained in the data, but destroy any coevolutionary couplings, frequently used in protein structure prediction. We find that phylogeny actually induces spurious non-zero couplings. These are, however, significantly smaller that the largest couplings derived from natural sequences, and therefore have only little influence on the first predicted contacts. However, in the range of intermediate couplings, they may lead to statistically significant effects. Dissecting phylogenetic from functional couplings might therefore extend the range of accurately predicted structural contacts down to smaller coupling strengths than those currently used.

## Introduction

Global coevolutionary modeling approaches have recently seen a lot of interest [1, 2], either directly for predicting residue-residue contacts from sequence ensembles corresponding to homologous protein families [3–5], in predicting mutational effects [6–8], or even in designing artificial but functional protein sequences [9–12], or as an input to deep-learning based protein structure prediction. The latter approach has recently lead to a breakthrough in predicting protein structure from sequence [13–17].

The basic idea of coevolutionary models, like the Direct-Coupling Analysis (DCA) [6], is that the amino-acid sequences, typically given in the form of a multiple-sequence alignment (MSA) of width (or aligned sequence length) $L$, can be considered as a sample drawn from some unknown probability distribution $P(a_1, \ldots, a_L)$, with $(a_1, \ldots, a_L)$ being an aligned amino-acid sequence. This probabilistic model is typically parameterized as $P(a_1, \ldots, a_L) \propto \exp\{\Sigma_{i<j} J_{ij}(a_i, a_j) + h_i(a_i)\}$ via biases (or fields) $h_i(a_i)$ representing site-specificities in amino-acid usage (i.e. patterns of amino-acid conservation), and via statistical couplings $J_{ij}(a_i, a_j)$, which represent coevolutionary constraints and cause correlated amino-acid usage in positions $i$ and $j$ [18].

In most cases, the parameters of these models are inferred by (approximate) maximum-likelihood, under the assumption that the sequences in the MSA are (almost) independently and identically distributed according to $P(a_1, \ldots, a_L)$. On one hand, this assumption is needed to make model inference from MSA technically feasible. On the other hand, it is in obvious contradiction to the fact that sequences in homologous protein families share common ancestry in evolution, and therefore typically show considerable phylogenetic correlations, which can be used to infer this unknown ancestry from data [19]. Phylogeny induces highly non-trivial correlations between MSA columns [20], which however do not represent any functional relationship.

Disentangling correlations induced by functional or structural couplings from phylogeny-caused correlations turns out to be a highly non-trivial task [20–22]. Simple statistical corrections have been proposed, like down-weighting similar sequences when determining statistical correlations [3], or the average-product correction (APC) [23] applied to the final coevolutionary coupling scores. While sequence weighting has initially been reported to significantly improve contact prediction, recent works show little effect [24], probably due to the fact that, e.g., Pfam [25] is now based on reference proteomes and therefore less redundant than databases used to be about a decade ago. APC was shown to be more of a correction of biases related to amino-acid conservation than to phylogeny [26].

To make progress, we suggest a complementary approach. Instead of removing phylogenetic correlations from DCA-type analyses, we suggest null models having the same

conservation and phylogenetic patterns of the original MSA of the protein family under consideration, but strictly lack any functional or structural couplings.

Running DCA on artificial MSA generated by these null models, and comparing them to the results obtained from natural MSA, we find some remarkable results: while the largest eigenvalues of the residue-residue covariance matrix appears to be dominated by phylogenetic effects, the strongest DCA couplings are hardly influenced by phylogeny. The spurious couplings induced by phylogeny are, however, non-zero, and may limit the accuracy of contact prediction when going beyond the first few strongest couplings. More precisely, our observations are quantitatively compatible with the idea, that already the first false-positive contact prediction are caused by not correctly treating phylogeny in DCA. This shows also that methods properly dissecting phylogenetic and functional correlations in sequence data have a high potential to substantially extend the contact predictions beyond current methods.

The paper is organized as follows. After this introduction, we provide the *Materials and Methods*, with a short review of DCA, but most importantly with the presentation of three null models. The *Results* section compares the spectral properties of the residue-residue covariance matrix of the real data with those of MSA generated by the null models, followed by an assessment of the couplings inferred by DCA and their relation to residue-residue contacts. The *Conclusion* sums up the results and discusses potentially interesting future directions. Supporting tables and figures are shown in S1 Text.

## Materials and methods

### Protein families, sequence alignments and Direct Coupling Analysis

Coevolutionary analysis is mostly applied to families of homologous proteins (or protein domains), as provided by the Pfam database [25]. Multiple sequence alignments (MSA) can be downloaded in the form of rectangular arrays $\mathbf{D} = \{a_i^m \mid i = 1, \ldots, L; \ m = 1, \ldots, M\}$ of width $L$ (aligned sequence length) and depth $M$ (number of aligned sequences). The entries $a_i^m \in \{-, A, C, \ldots, Y\}$ are either one of the 20 standard amino acids, or alignment gaps represented as "−". Note that insertions are not aligned in Pfam alignments, and are therefore typical removed from the MSA before statistical model learning. Here $L$ describes the sequence length after removal of insertions. For the structural analysis, the Pfam MSA is mapped to experimentally resolved PDB protein structures [27], and distances are measured as minimum distances between heavy atoms. Following established standards in the coevolutionary literature, we use a cutoff of 8Å for residue-residue contacts.

For our work, we have selected three datasets:

- **DS1**: Detailed results are given for 9 Pfam protein families, with not too long sequences ($L < 250$), not too large MSA ($M < 10,000$ after removing duplicate sequences) and available PDB structures, see details in Table A in S1 Text.

- **DS2**: Statistical results are given for 60 Pfam protein families belonging to the PSICOV benchmark set [28]. Only families with $M < 12,000$ pairwise distinct sequences were considered, cf. Table B in S1 Text.

- **DS3**: We have also compiled and dataset of 20 smaller Pfam protein families with known PDB structures, to study the influence of finite MSA depth on our findings, cf. Table C in S1 Text.

Global coevolutionary models, as those constructed by DCA, describe the sequence variability between the members of a protein family, i.e. between different rows of the MSA $\mathbf{D}$, via

a statistical model

$$P(a_1, \ldots, a_L | \mathbf{J}, \mathbf{h}) = \frac{1}{Z} \exp \left\{ \sum_{1 \leq i < j \leq L} J_{ij}(a_i, a_j) + \sum_{1 \leq i \leq L} h_i(a_i) \right\} \ , \qquad (1)$$

parameterized via pairwise coevolutionary residue-residue couplings $J_{ij}(a_i, a_j)$ and single-residue biases (or fields) $h_i(a_i)$, while $Z$ is a normalization factor also known as partition function. In the simplest setting, these parameters are inferred from the data via maximum likelihood, i.e.

$$\{\mathbf{J}, \mathbf{h}\} = \mathrm{argmax}_{\mathbf{J}, \mathbf{h}} \prod_{m=1}^{M} P(a_1^m, \ldots, a_L^m | \mathbf{J}, \mathbf{h}) \ . \qquad (2)$$

This maximization leads directly to the fact, that the model $P$ reproduces the empirical statistics of single MSA columns and of column pairs,

$$
\begin{aligned}
f_i(a_i) &= \sum_{\{a_k | k \neq i\}} P(a_1, \ldots, a_L) \ , \\
f_{ij}(a_i, a_j) &= \sum_{\{a_k | k \neq i, j\}} P(a_1, \ldots, a_L) \ ,
\end{aligned}
\qquad (3)
$$

where $f_i(a)$ represents the fraction of amino acids $a$ in column $i$, i.e. the *residue-conservation* statistics, while the $f_{ij}(a, b)$ describe the fraction of sequences having simultaneously amino acids $a$ and $b$ in columns $i$ and $j$, thereby representing *residue covariation / coevolution*, i.e. the correlated usage of amino acids in pairs of columns, cf. Fig 1. The inference of the parameters is a computationally hard task, since, e.g., the computation of the marginals in Eq (3) depends on an exponential sum over $\mathcal{O}(21^L)$ terms. Many approximation schemes have been proposed, we use plmDCA [29] based on pseudo-likelihood maximization, since it represents a well-tested compromise between accuracy and running time.

A particularity of this approach is that Eq (2) assumes that the sequences in the MSA **D** form an independently and identically distributed sample of $P(a_1, \ldots, a_L)$ and that the likelihood can be factorized into a product over the rows of **D**. This assumption is incorrect; biological sequences are the result of natural evolution and thus show hierarchical phylogenetic relations. Phylogeny by itself leads to a non-trivial correlation structure between different residue positions with a power-law spectrum [20], and this leads to non-zero but also non-functional residue-residue couplings when using DCA. These may interfere with the functional couplings, which are e.g. used for residue-residue contact prediction from MSA data, and thereby negatively impact prediction accuracy.

It is notoriously hard to disentangle the two, cf. [21, 22]. The problem is that evolution is a non-equilibrium stochastic process, whose dynamics in principle depends on the evolutionary constraints represented, e.g., by the couplings and fields in the DCA model. Global model inference from phylogenetically correlated data remains an open questions.

Here we follow a different route. We define different null models, which take residue conservation and in part also phylogeny into account, but do not show any intrinsic amino-acid covariation. The null models allow us to create large numbers of suitably randomized sequence ensembles, on which standard DCA can be run. The couplings resulting from randomized data can be used to assess the statistical significance of the couplings resulting from the real MSA **D**, and therefore to discard purely phylogeny-caused couplings. While, to the best of our knowledge, this has never been done in the context of protein families and DCA, somehow

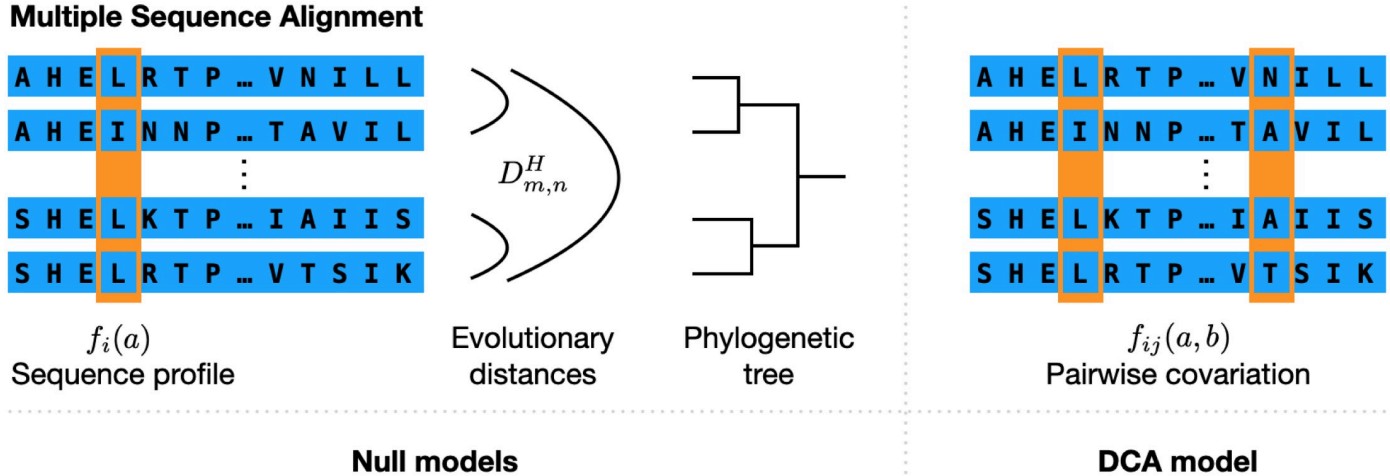

**Fig 1. Schematic representation of the information used by DCA and the null models.** MSA contain several types of information about the sequence variability. The sequence profile and residue covariation describe the statistics of individual MSA columns and column pairs, both are used in DCA. However, the MSA contains also phylogenetic information, here represented by the matrix $\{D_{mn}^{H}|1 \leq m < n \leq M\}$ of Hamming distances between sequences, or by the (inferred) phylogenetic tree. The different null models use the profile and phylogenetic information, but no residue covariation.

similar techniques have been proposed in the context of phylogenetic profiling [30], but applied to correlations rather than couplings.

### Null model I: Profile-aware sequence randomization

The first null model is very simple. It randomizes the input MSA by conserving the single-column statistics $f_i(a)$, for all sites $i = 1, \ldots, L$ and all amino acids or gaps $a \in \{-, A, \ldots, Y\}$. This is done by simple random but independent permutations of all MSA columns. This destroys all correlations between positions (the coevolutionary ones) and between sequences (the phylogenetic ones), only the residue conservation patterns of the original MSA are preserved. Formally, the randomized sequences become an independently and identically distributed sample from the *profile model*

$$P_{profile}(a_1, \ldots, a_L) = \prod_{i=1}^{L} f_i(a_i) \ . \tag{4}$$

So in principle there are no couplings between different residues at all. However, when running DCA on this sample, inferred couplings will be non-zero due to finite sample size. They will take distinct values from one randomization to the next, but there may be systematic biases due to the distinct conservation levels, which are maintained as compared to the original Pfam MSA.

### Null model II: Profile- and phylogeny-aware sequence randomization

The second null model is more complicated, since it preserves also (at least approximately) the phylogenetic information contained in the original MSA. Here we assume that this information is coded in the pairwise distances between sequences, i.e. in the matrix $\{D_{mn}^{H}|1 \leq m < n \leq M\}$ of Hamming distances between all pairs of sequences, as is done in distance-based phylogeny reconstruction [19, 31].

The aim of the second null model is to construct a randomized MSA which preserves both the sequence profile given by the position-specific frequencies $f_i(a)$, and the matrix $\{D_{mn}^{H}\}$ of

pairwise Hamming distances between sequences. This can be achieved by the following Markov chain Monte Carlo (MCMC) procedure acting on the entire alignment. Our method is initialized using a sample of null model I, i.e. all coevolutionary and phylogenetic information from the original MSA is destroyed, but the profile is preserved. The resulting randomized MSA after $t$ MCMC steps is called $\tilde{\mathbf{D}}_t = \{\tilde{a}_i^m \mid i = 1, \ldots, L, \; m = 1, \ldots, M\}$.

In step $t \to t+1$, one column $i \in \{1, \ldots, L\}$ is selected randomly, as well as two rows $m, n \in \{1, \ldots, M\}$. An exchange of the entries $\tilde{a}_i^m$ and $\tilde{a}_i^n$ is attempted, to obtain a new matrix $\tilde{\mathbf{D}}_{t+1}$. This matrix is accepted with a Metropolis-Hastings acceptance probability $p_{acc}$ of

$$p_{acc} = \min[1, \exp\{-\beta(\|D^H(\tilde{\mathbf{D}}_{t+1}) - D^H(\mathbf{D})\| - \|D^H(\tilde{\mathbf{D}}_t) - D^H(\mathbf{D})\|)\}] \; , \tag{5}$$

otherwise the exchange is refused and the matrix $\tilde{\mathbf{D}}_t$ remains invariant in step $t+1$. In this expression, $D^H(\mathbf{D})$ stands for the matrix of Hamming distances between the rows of the original MSA $\mathbf{D}$, analogously for the randomized MSA, and $\|\cdot\|$ for the Frobenius norm of matrices. This acceptance rule guarantees that each exchange, which brings the distance matrix $D^H(\tilde{\mathbf{D}}_t)$ closer to the target matrix $D^H(\mathbf{D})$, is accepted. Exchanges going into the opposite direction are accepted with a smaller probability depending exponentially on the formal "inverse temperature" $\beta$. Here we use simulated annealing, i.e. we initialize $\beta$ in a small value and slowly increase it over time, in order to force the Hamming distances of the randomized MSA closer an closer to the ones of the natural MSA. Fig A in S1 Text illustrates that very high correlations (Pearson correlation $> 0.97$) between the two distances matrices are actually obtained across protein families by our algorithm. Fig B in S1 Text shows the distance histograms between natural and randomized sequences. We find that, with rare exceptions, randomized sequences are at most at 60–70% sequence identity (minimal Hamming distance 30–40%) to the closest natural sequences, showing that Null model II actually generates sequences, which are distant from the original Pfam MSA.

Since the sequence profile remains unchanged by this procedure, and the natural distances between sequences are approached, the randomized MSA thus contains approximately the same conservation and phylogenetic properties of the biological sequence data. However, potentially existing functional correlations between MSA columns are eliminated. The resulting data-covariance matrix is expected to have non-trivial properties in agreement with [20], and DCA is expected to be able to reproduce this correlation structure via couplings $J_{ij}(a, b)$. Repeating the randomization many times, we can assess the statistics of phylogeny-generated couplings, and thereby the significance of the couplings found by running DCA on the original protein sequences collected in $\mathbf{D}$.

Note also that, in the limit where the formal inverse temperature in Eq (5) is set to $\beta = 0$, i.e. in the case of infinite formal temperature $T = \beta^{-1}$, we recover Null model I. One could use $\beta$ therefore as an interpolating parameter between these two Null models.

## Null model III: Profile- and phylogeny-aware sequence resampling

To corroborate the results of Null model II, we have also used a complementary strategy using explicitly an evolutionary model and a phylogenetic tree to resample sequences on this tree. The evolutionary model we use is the Felsenstein model for independent-site evolution [32], i.e. a model which does account for site-specific conservation profiles and phylogeny, but not for any intrinsic correlation / coupling between different sites. In this context, the stationary probability distribution of sequences is described by a profile model

$$P_\omega(a_1, \ldots, a_L) = \prod_{i=1}^{L} \omega_i(a_i) \; , \tag{6}$$

which has the same form of the profile model in Eq (4), but the factors are not given directly by the empirical amino-acid frequencies in the MSA columns. All sites $i = 1, \ldots, L$ evolve independently, and for each site $i$ the probability of finding some amino acid $b$, given an ancestral amino acid $a$ some time $t$ before, is given by

$$P(a_i = b \,|\, a_i = a, t) = e^{-\mu t} \delta_{a,b} + (1 - e^{-\mu t}) \omega_i(b) \ , \tag{7}$$

with $\mu$ being the mutation rate and $\delta_{a,b}$ the Kronecker symbol, which equals one if and only if the two arguments are equal, and zero else. In this model, there is no mutation with probability $e^{-\mu t}$, and the amino acid in position $i$ does not change, or at least one mutation with probability $1 - e^{-\mu t}$. In the latter case, the new amino acid $b$ is emitted with its equilibrium probability $\omega_i(b)$. While being simple, the Felsenstein model of evolution is frequently used in phylogenetic inference.

The algorithm proceeds in the following way, using the implementation of [22]:

- A phylogenetic tree $\mathcal{T}$ is inferred from the MSA **D** using FastTree [33]. Instead of using inter-sequence Hamming distances (like in Null model II), FastTree is using a maximum-likelihood approach based on a model of independent-site evolution, i.e. no coevolutionary information is taken into acocunt in tree inference.

- The mutation rate $\mu$ and all site-specific frequencies $\{\omega_i(a)\}$ are inferred using maximum likelihood.

- To resample the MSA according to this model, the root sequence is drawn randomly from $P_\omega$, and stochastically evolved on the branches of $\mathcal{T}$ using the transition probability Eq (7).

- The resampled MSA is composed by the sequences resulting in the leaves of $\mathcal{T}$.

This procedure allows thus to emit many artificial MSA being evolved on the same phylogeny and with the same stationary sequence distribution as the one inferred from the natural sequences given in **D**, but no coevolutionary information is taken into account at any of the four steps. Note that the emitted MSA are expected to be more noisy than the ones of Null model II. In particular the column statistics will differ from $f_i(a)$, and also the inter-protein Hamming distances $D^H$ will differ more from the ones in the training MSA, cf. Fig C in S1 Text showing that correlations between the $D^H$ matrices remain large but not as large as in Null model II (Pearson correlations 0.7–0.95 for the protein families in dataset DS1).

Again DCA can be run on many of the resampled MSA, and the DCA couplings of the natural MSA can be compared with the statistics of the resampled ones, to assess their statistical significance beyond finite-size and phylogenetic effects.

## Results and discussion

The two Null models II and III, which both include phylogenetic correlations between proteins, lead to qualitatively coherent, but quantitatively slightly different results, which reflect the different randomization strategies. In the main text of this article, we will present almost exclusively the results of Null model II, in comparison to the natural MSA and Null model I. The results for Null model III are delegated to S1 Text, unless explicitly stated.

### The spectral properties of the residue-residue correlation matrix are dominated by phylogenetic effects

Following the mathematical derivations published in [20], we would expect that the residue-residue covariance matrix $\mathbf{C} = \{c_{ij}(a, b) \,|\, i, j = 1, \ldots, L; a, b \in \{-, A, \ldots, Y\}\}$ with $c_{ij}(a,$

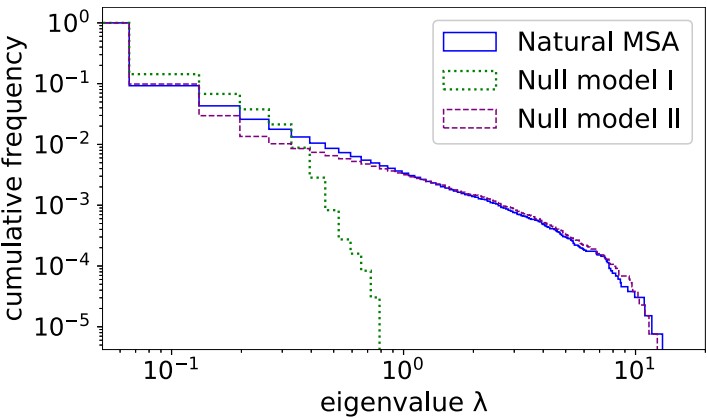

**Fig 2. Eigenvalue spectra of the covariance matrix of the natural MSA and for Null models I and II.** We show the cumulative distribution of the unified eigenvalue spectra for the 60-protein dataset DS2, i.e. the fraction of eigenvalues larger than λ is shown as a function of λ. We observe that the phylogeny-aware Null model II shows the same fat tail for large eigenvalues, which is also present in the natural data, while the non-phylogenetic Null model I has a more compact support. The cutoff of the tail for large λ is an effect of the inter-family variability of the largest eigenvalues among the 60 spectra, cf. Fig D in S1 Text for the 9 individual proteins in dataset DS1.

$b) − f_i(a)f_j(b)$ is strongly impacted by phylogenetic correlations in the data. More precisely, while totally random data would lead to the Marchenkov-Pastur distribution for the eigenvalue spectrum of **C**, the hierarchical structure of data on the leaves of a phylogenetic tree leads to a power-law tail of large eigenvalues.

It is thus not very astonishing, that both Null models II and III show fat tails in the spectrum of their data covariance matrices **C** (even if Null model II does not fulfill the mathematical conditions of the derivation in [20] because not generated according to a hierarchical process), while the spectrum of Null model I has a substantially more compact support, cf. Fig 2, and Figs D and E in S1 Text. The interesting observation is that, at the level of the eigenvalue spectrum, the natural data are hardly distinguishable from the phylogeny-aware Null models II and III, in difference to Null model I.

This suggests the following conclusions: the dominant global residue-residue correlation structure, as far as reflected by the largest eigenvalues of the **C**-matrix, results from phylogeny. A comparison with principal-component analysis (PCA) relates these eigenvalues to the large-scale organization of sequences in sequence space, e.g. into clusters of sequences. Note that the eigenvectors are expected to contain complementary information, e.g. used for PCA or for the identifcation of protein sectors [34, 35], defined as multi-residue groups of coherent evolution.

## Phylogenetic effects induce couplings in DCA, but these are smaller than couplings found in natural sequences

However, the couplings derived by DCA are not directly related to the largest eigenvalues of the residue-residue covariance matrix. Actually, the computationally most efficient DCA approximations based on mean-field [3] or Gaussian [36] approximations, relate the couplings **J** to the negative of the inverse of **C**. The DCA couplings are therefore dominated by the smallest eigenvalues of **C**, cf. also [37].

Here we use plmDCA, the resulting couplings therefore lack any simple relation to the eigenvalues and eigenvectors of the residue-residue covariance matrix. In difference to standard plmDCA we do not use any sequence weighting, since it might interfere with the

phylogenetic signal in a non-controlled way. In Fig 3, we plot histograms of the DCA couplings (APC-corrected Frobenius norms $F^{APC}$ of the coupling matrices for each residue pair, i.e. the standard output of plmDCA, cf. [29] and the Introduction for a short explanation of APC) for Null models I, II and the natural MSA **D** for the datasets DS2 of large and DS3 of small-medium depth MSA. Equivalent results for the nine individual families in DS1 are shown in Fig F in S1 Text, along with those for Null model III in Fig G in S1 Text.

We see that across all protein families, DCA couplings from natural data reach significantly larger values than those derived from both Null models. The latter two miss in particular the strong tail for large values; their supports being much more concentrated. For large MSA (DS2), the phylogeny-aware Null model II generates larger couplings than the phylogeny-unaware Null model I, i.e. they go beyond what is to be expected from finite-sample effects alone. This latter difference almost vanishes for smaller MSA (DS3), where the coupling histograms for Null models I and II almost coincide. Interestingly, the phylogenetic couplings of Null model II are almost invariant with respect to family size, while finite-size effects (Null model I) decrease with family size, and the tail of large couplings resulting from natural sequences grows with family size.

It is very interesting that, while the spectra are similar for natural data and phylogeny-aware null models, the dominant residue-residue couplings are neither explainable by phylogeny nor by finite sample size. They must consequently result from intrinsic evolutionary constraints acting on the proteins due to natural selection for correctly folded and properly functioning proteins.

## Residue-residue contact predictions are moderately impacted by phylogenetic effects

This observation becomes even more interesting, when we compare the couplings of residue-residue contacts and non-contacts. In Fig 4A, we show normalized coupling histograms for the two subsets of residue pairs in dataset DS2. The tail of large couplings is present only in the contacts, explaining why DCA and the closely related GREMLIN accurately predict contacts when couplings are high enough [38, 39], cf. also the positive predictive value (PPV) in function of the coupling in Fig 3. As is visible in the Fig H in S1 Text for the nine individual protein families, the contact prediction in a family depends crucially on the size of this tail of large couplings.

Using Null model II, we cann assess the strength of phylogeny-induced spurious couplings on the same subsets of contacts and non-contacts extracted for DS2, cf. Fig 4B and Fig I in S1 Text (and similarly Fig J in S1 Text for Null model III). We see that the two histograms for contacts and non-contacts get almost identical to each other; due to the larger number of non-contacts the largest couplings are therefore dominated by non-contacts across all studied protein families. Most interestingly, when we compare the non-contact histograms in both panels of Fig 4, they are extremely similar. It appears that the strength of the non-contact couplings in the natural data is mostly consistent with the phylogeny-induced spurious couplings in Null model II.

The differences between these histograms translates into differences between PPV-curves as shown in Fig 5. The upper panels show the results for the union of all predictions, i.e. the largest DCA-couplings scores $F^{APC}$ of all families come first, for the dataset DS2 of large MSA (Panel A) and the DS3 of smaller MSA (Panel B). We see that the largest couplings derived from the natural MSA are contacts in both cases, but much more contacts are found in the larger MSA. However, in the lower panels we see that for smaller MSA only about 60% of the considered families have such large scores, leading to an initial average PPV (each family

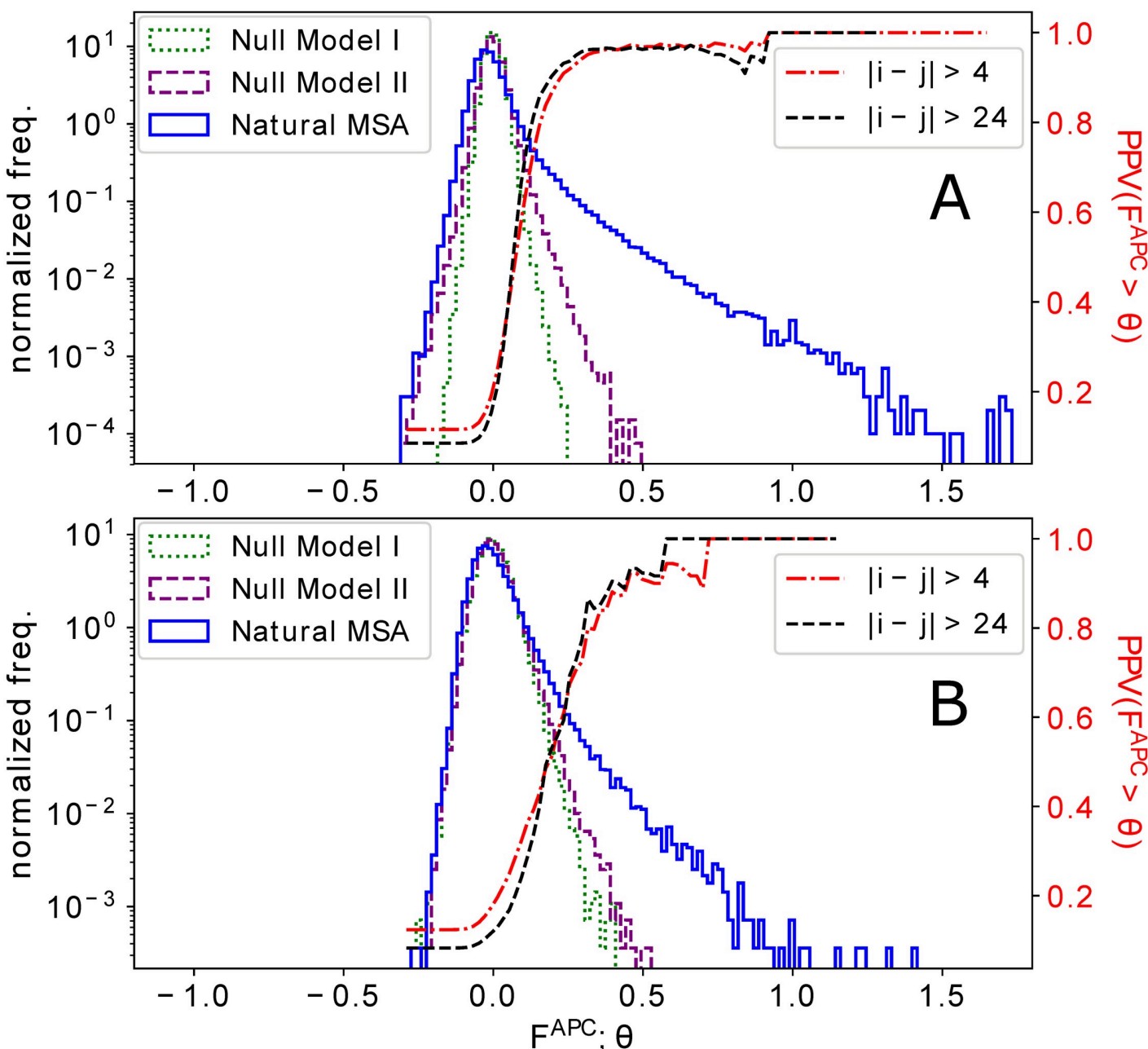

**Fig 3. DCA scores derived from natural sequence data and from MSA generated by Null models I and II, for datasets DS2 (panel A) of large MSA, and DS3 (panel B) of smaller MSA.** For the protein families under study, we show the histograms of DCA coupling scores $F^{APC}$ (APC corrected Frobenius norm of couplings, the standard output of plmDCA), for the natural MSA and samples of Null models I and II. Here and in the following, histograms are normalized as probability distributions, *i.e.* to area one under the curve. It becomes evident that phylogenetic effects create—at least for sufficiently deep MSA—larger couplings than to be expected from finite sample size alone. However, couplings derived from the natural MSA have substantially larger values. The figures include also the positive predictive value (PPV, scale on the right of each panel), providing the fraction of true contacts in between all couplings $F^{APC}$ above some threshold $\theta$, as a function of $\theta$, for plmDCA run on the natural MSA. We clearly see that almost all large couplings correctly predict contacts, while the PPV starts to drop once we reach $F^{APC}$ reached also by phylogenetic effects in Null model II. We find this to be true for all non-trivial contacts (sequence separation $|i - j| > 4$) as well as for long-distance contacts ($|i - j| > 24$).

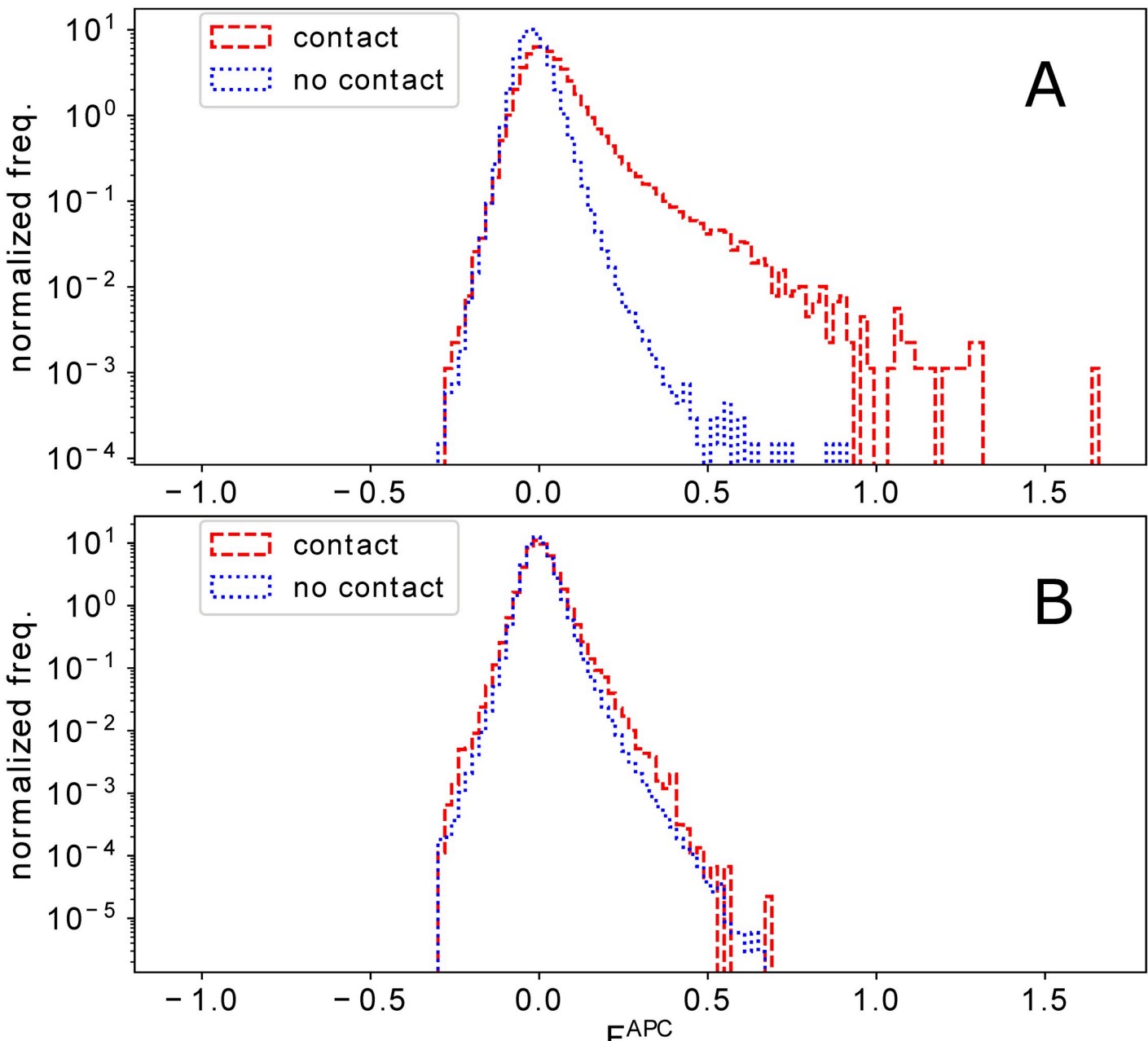

**Fig 4. Histogram of DCA scores derived from natural sequence data (Panel A) and Null model II (Panel B) for residue-residue contacts and non-contacts.** For the protein families in DS2, we show the histograms of DCA coupling scores (APC corrected Frobenius norm of couplings), separated for contacts and non-contacts (defined using the representative protein structures in Table B in S1 Text). Only pairs with linear separation $|i - j| > 4$ along the chain are taken into account.

considered individually, and the individual PPV are averaged) of only about 0.6, while almost all large MSA lead to initially accurate contact predictions. The figures also contain a contact prediction for randomized data from Null model II. In the upper panels we observe a very weak contact signal; it results from the fact that conserved sites tend to lead to larger phylogeny-induced spurious couplings, but they also tend to be concentrated in proteins, e.g. in active sites or the protein core, and consequently to have a higher contact fraction.

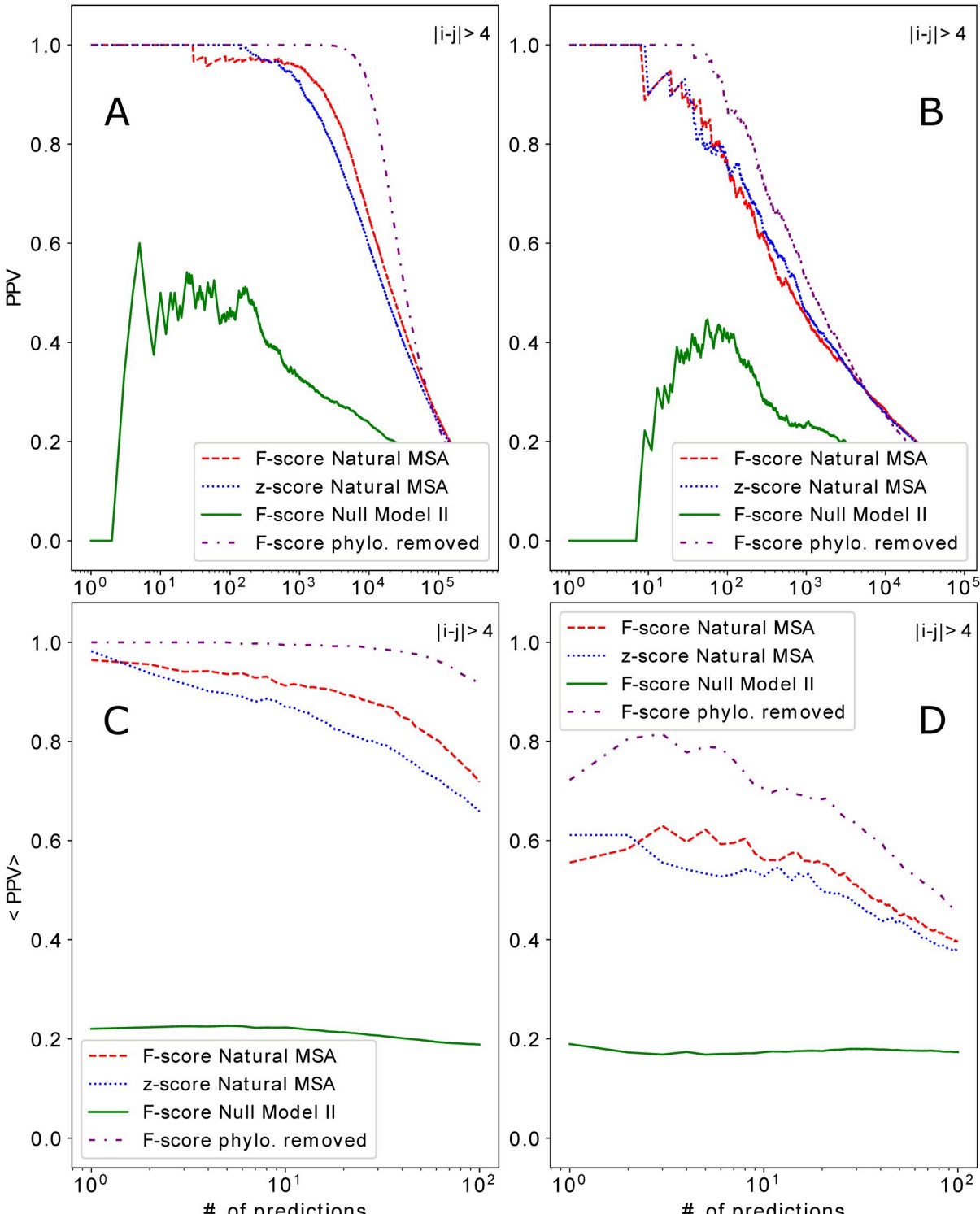

**Fig 5. PPV for residue-residue contact prediction from natural data and Null model II.** The positive predictive values for residue contact prediction are shown for datasets DS2 (Panels A and C) and DS3 (Panels B and D), using the natural data (red, blue) and randomized data from Null model II (green). The upper panels (A,B) show joint contact prediction for all proteins, the lower panels (C,D) the averages over the individual PPV curves for all single families. All panels show also *hypothetical* PPV curves (purple), which might be reached by a method removing phylogenetic biases; they artificially combine DCA scores obtained from natural MSA on contacts, and from Null model I on non-contacts.

The histograms in Figs 3 and 4 are derived from individual samples of the Null models. One might expect that they change from sample to sample. While this is the case for individual couplings, the histograms remain remarkably unchanged when comparing samples, cf. Figs K and L in S1 Text. These observations show us that, while phylogenetic effects result in non-zero couplings between residues when DCA is applied, these couplings are relatively weak and never reach the size of the couplings, which allow for a high-confidence contact prediction. This idea is also corroborated by the quantitative assessment of the statistical significance in the couplings derived from natural sequences, as compared to the ones generated by the Null models. To this aim, we assign a z-score to each residue pair $(i, j)$: Using 50 samples of Null model II, we determine the mean and standard deviation of couplings derived from Null model II, individually for each pair $(i, j)$. We use these values to determine the z-score, i.e. the number of standard deviations, the actual couplings (from natural MSA) is away from the means for Null model II. In Fig 6, we observe, that this z-score is highly correlated with the plmDCA score derived from natural MSA, across all families. Almost all DCA scores above 0.2–0.3 have highly significant z-scores above 3 or even more. Even larger correlations between DCA and z-scores are observed in Null model III, cf. Fig M in S1 Text. One might be tempted to use this statistical significance score instead of the DCA-coupling strength for contact

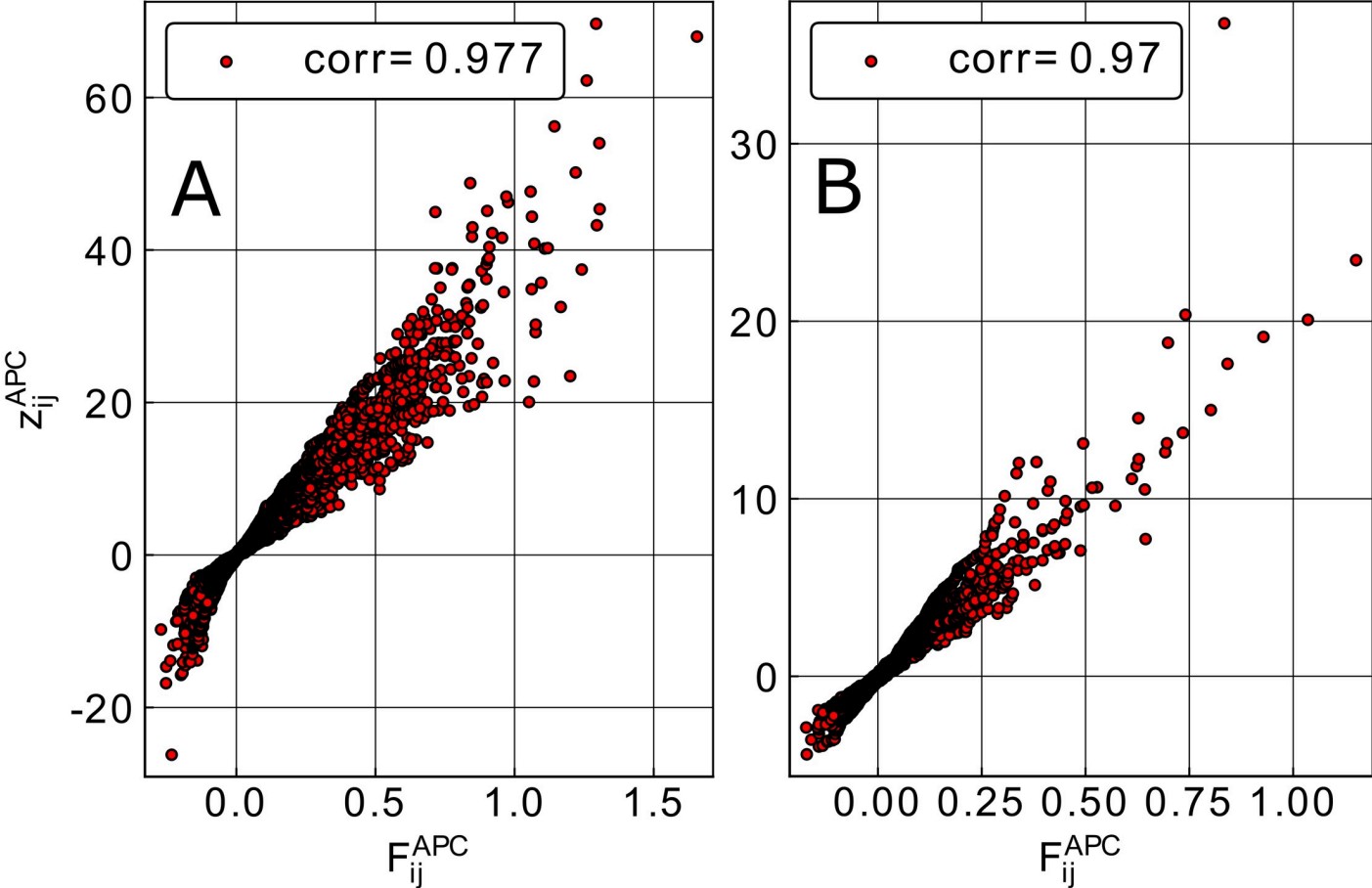

**Fig 6. z-scores of couplings derived from the natural MSA, as compared to the distribution of couplings derived from Null model II.** For each residue pair $(i, j)$, we calculate the z-score for the DCA score derived from natural data as compared to 50 realizations of Null model II. Panel A shows the data for the dataset DS2 of large MSA, Panel B for DS3 of small-intermediate MSA.

prediction. In Fig 5 we show that the two lead to highly comparable results, with a slight advantage for the standard $F^{APC}$ after the first few predictions. This difference might result from the before-mentioned observation that conserved positions tend to have larger phylogeny-induced couplings (and thus larger variances between different samples of Null model II), causing a systematic reduction of the related z-scores.

## Conclusion

Global coevolutionary modeling treats multiple-sequence alignments of homologous protein sequences as collections of independently and identically distributed samples of some unknown probability distribution $P(a_1, \ldots, a_L)$, which has to be reconstructed from data. The assumption of independence is obviously violated due to the common evolutionary history, in particular sequences from related species show strong phylogenetic correlations.

It is, however, notoriously difficult to unify the idea of a global model including coevolutionary covariation between sites and phylogenetic correlations between sequences. Statistical corrections may improve the situation slightly, but they are too simple to take the hierarchical correlation structure into account, which is generated by the evolutionary dynamics on a phylogenetic tree.

Here we have proposed to approach this problem in a complementary way, by introducing null models—i.e. randomized or re-emitted multiple-sequence alignments—which reproduce conservation and phylogeny, but do not contain any real coevolutionary signal. When applying Direct Coupling Analysis as a prototypical global coevolutionary modeling approach, we observe that phylogenetic correlations between sequences lead to a changed residue-residue correlation structure, represented by a fat tail in the eigenvalue spectrum of the data covariance matrix. It leads also to distributed couplings, which, however, are smaller than the largest couplings found when applying DCA to natural sequence data, i.e. smaller than the couplings used for residue-residue contact prediction. The latter are significantly larger than couplings resulting from phylogeny, i.e. we can conclude that the first predicted contacts are influenced only to a very limited degree by phylogenetic couplings.

However, it is also striking that, across the studies protein families, the phylogeny-caused couplings in Null models II and III almost reach the DCA-score threshold found before for accurate contact prediction. This suggests that the suppression of phylogenetic biases in the data (or their better consideration in model inference), may shift this threshold down and therefore allow for predicting much more contacts. The potential gain would be limited due to the finite depths of the MSA, whose effects are assessed by Null model I. We can therefore quantitatively assess the potential in removing phylogenetic effects by the following *hypothetical* DCA output: on all contacts in our dataset DS2 and DS3 we use the standard plmDCA scores derived from natural data, and on all non-contacts we remove phylogenetic effects by using couplings derived by running plmDCA on samples of Null model I. The resulting *hypothetical* PPV curves are given in Fig 5 as purple lines: they are substantially higher than the real PPV obtained from the original data. In the case of the large MSA in DS2, we find a broad plateau of almost perfect PPV close to one, starting to drop only after about 50 top residue pairs, but staying above 90% even after 100 pairs, as compared to about 70% for the real DCA predictions. Even in the small-to-medium-depth MSA of DS3 the potential effect of removing phylogenetic effects is considerable, even if the finite-sample effect is much more pronounced. While in the real data only less than 60% of the considered protein start with a true-positive prediction, the hypothetical phylogeny-removed prediction starts with a PPV above 70%. Since coevolution-based scores are also input to most of the recent deep-learning-based

contact predictors, we could imagine that corrections for phylogenetic effects would also improve the accuracy of these methods.

Using many realization of the Null models, we can provide a z-score for the couplings found in the original sequence data, and thereby assess their statistical significance beyond effects of finite and phylogenetically correlated sampling. This is of interest in exploratory studies: in a recent study, one of us has used Null model II in a collaboration aiming at finding potential epistatic effects in a global analysis of more than 50,000 SARS-Cov-2 genomes [40]. Due to the obvious strong correlation between these very recently diverged genomes, potential epistatic couplings have to be assessed carefully, and scoring them by Null model II has turned out to be an essential element in the identification of a sparse, but statistical significant genome-wide network of epistatic couplings.

## Supporting information

**S1 Text. The file contains all supporting tables and figures cited in the main text.** (PDF)

## Acknowledgments

We thanks Pierre Barrat-Charlaix and Alejandro Lage-Castellanos for numerous discussions.

## Author Contributions

**Conceptualization:** Martin Weigt.

**Data curation:** Edwin Rodriguez Horta.

**Investigation:** Edwin Rodriguez Horta, Martin Weigt.

**Methodology:** Edwin Rodriguez Horta, Martin Weigt.

**Software:** Edwin Rodriguez Horta.

**Supervision:** Martin Weigt.

**Validation:** Edwin Rodriguez Horta.

**Visualization:** Edwin Rodriguez Horta, Martin Weigt.

**Writing – original draft:** Edwin Rodriguez Horta, Martin Weigt.

**Writing – review & editing:** Martin Weigt.

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
