## [Decision Letter · Decision Letter 0]

7 Oct 2020

Dear Weigt,

Thank you very much for submitting your manuscript "Phylogenetic correlations have limited effect on coevolution-based contact prediction in proteins" for consideration at PLOS Computational Biology.

As with all papers reviewed by the journal, your manuscript was reviewed by members of the editorial board and by several independent reviewers. In light of the reviews (below this email), we would like to invite the resubmission of a significantly-revised version that takes into account the reviewers' comments.

Please consider answering all the criticisms

We cannot make any decision about publication until we have seen the revised manuscript and your response to the reviewers' comments. Your revised manuscript is also likely to be sent to reviewers for further evaluation. In particular you need to show that the conclusions is also of relevant in more families, in particular in much smaller ones and that it really helps improving predictions.

Sincerely,

Rita Casadio

Guest Editor

PLOS Computational Biology

Arne Elofsson

Deputy Editor

PLOS Computational Biology

Please consider answering all the criticisms

Reviewer's Responses to Questions

**Comments to the Authors:**

Reviewer #1: Coevolution analysis forms the basis of many recent methods for template-free protein structure prediction. It aims to predict 3D contacts between residues in protein domains from their statistical signature of coevolution. All methods in use today treat the sequences in the input multiple sequence alignment (MSA) as being independent, with some simple sequence weighting to correct for different degrees of redundancy. This is a gross assumption because in fact the protein sequences are statistically dependent on each other, as described by the phylogenetic tree that can be reconstructed from the MSA.

This study investigates the influence of the assumption of independence by randomizing real MSAs such that the sequence composition at each position is conserved and the pairwise distances between the sequences is conserved, while the statistical signature of co-evolution signatures between residues is remove (except for noise that remains, of course).

The main results of the study are reported to be

(1) that spurious co-evolution signatures caused by the unjustified approximation of independence are not limiting our ability to predict contacts for the most strongly co-evolving pairs of positions;

(2) that we could predict many more contacts if we were able to suppress the spurious couplings by a proper treatment of the phylogenetic dependence of the sequences.

Main issues:

1. Result (1) is not as general as the authors purport. It only applies to MSAs with very many sequences and as such is close to self-fulfilling prophecy. If the MSAs are large enough, the phylogenetic noise will not limit the contact predictions. Also, these results are not new and not surprising. We have known for a long time that the position pairs obtaining the top DCA scores in large MSAs are to a large fraction in contact, except for false positives caused by homo-multimerization and other well-described effects. Other studies have shown that the independence assumption adds considerable noise to the coupling scores.

2. To learn a huge number of coupling parameters without overfitting, the models such as GREMLIN and DCA use strong regularizing square penalties than push the coupling parameters towards zero. MSAs with many diverse sequences therefore produce higher scores, as the signal can better push against the strong regularizers. Therefore, the larger and more diverse the MSA is, the larger the top co-evolution scores usually are and the less limiting the influence of spurious signals from phylogeny will be. Therefore the depth and diversity of the MSA is an absolutely critical parameter to understand the influence and limitations induced by the independence assumption. The authors have ignored this parameter.

Also, they show results only for very large MSAs (table 1), large enough to predict with current methods a sufficient number of contacts reliably to predict their structures reliably. The interesting regime is of course that of an intermediate number of sequences, 50- 500 for example, for which it is still challenging to predict contacts.

3. So what? It is not clear in what way the insights in this study would help us to improve contact prediction in the future. How precisely might we reduce or remove the spurious signals from the unjustified assumption of independence of the sequences? How can we use the presented null models to improve contact prediction?

Reviewer #2: This is an interesting and insightful paper.

It tackles a long-standing issue in DCA, namely the relevance of phylogenetic correlations between sequences and how they affect the inferred co-evolutionary coupling strengths.

The authors address the problem elegantly, and their main result is that, mostly, phylogenetic correlations affect only low-strength couplings, so that high-strength ones can be safely trusted.

Although their work is convincing, I have some issues that I would like to see addressed.

1) As a practitioner, I (and I guess most people using DCA for practical applications) know that DCA works extremely well, with really very very few false positives within the first N contacts (N being the length of the sequences in the MSA). Once down-weighted by similarity, thus, it seems that phylogenetic correlations play only a minor role. This is perfectly in line with their findings.

My understanding is that correlated mutations for O(N) pairs give the same final scores whether they are "clustered" on branches of the tree or scattered everywhere. And in principle, the only reason why we should downplay their role when the same pair appears in a subbranch is not because they are close on the tree, but because the full sequences are close to each other (otherwise they would be wonderful co-occurrences to be duly appreciated by DCA). But this sequence similarity is precisely what re-weighting accounts for.

In this respect, I have to admit that I never felt the worry about phylogenetic correlations as compelling and urgent (interesting for sure to understand, but not necessarily as a major concern).

Could the authors comment?

2) Connected to this, I'd also like to comment on the null models they used in this manuscript, to understand to what extent theirs is not a circular argument.

My understanding is that phylogenetic trees are built using the independent site hypothesis. Some of them are built using (weighted) Hamming distances, others using likelihood approaches. In this respect, what the authors do is perfectly consistent with the correlation present in these trees. Nonetheless, that's the definition of phylogeny that people has given because i) it works and ii) it is mathematically and computationally tractable. Should the phylogenetic information that really affects DCA be encoded in ways that escape the independent site approximation used to build trees, it would not be addressed by the present manuscript.

In this respect they might more carefully phrase their results stating that phylogeny information related to tree-building is only mildly affecting the results, whereas deeper phylogenetic information, beyond the single site approximation, might still do it and this should be addressed in future work.

This is just a suggestion to stress that what is real (phylogenetic correlation) is different from the approximate ways we use to capture it (present tree-building algorithms).

Reviewer #3: In this paper authors performed an analysis based on Null models to support the idea that phylogenetic correlations only provide a limited contribution to coevolution-based contact prediction methods.

To prove their claim, authors defined two types of Null models preserving conservation only and conservation+phylogeny from the original MSA and compared DCA coupling obtained with these models against those derived from the original MSA. As result, they show that, while phylogenetic effects are dominant to determine the structure of the residue-residue correlation matrix, coupling derived by DCA using MSAs preserving conservation+phylogeny but not containing any coevolution signal are not sufficient to distinguish residue-residue contacts.

Overall, I believe that the procedure applied is sound and results are properly supported by data.

However, I suggest to better justify the choice of 9 protein families analyzed in this study, possibly providing more details on data selection. In general, I believe that enlarging the set of considered families would definitely add to the paper. Maybe, results for some additional family could be reported and discussed in the supplementary material.

Minor:

I suggest to reproduce all figures to improve readability in B/W

There is a missing citation in Fig4 caption

Reviewer #4: The authors investigate the influece of phylogenetic correlations in coevolutionary-based contact prediction.

The effect of phylogentic correlations is investigated by analyzing the performances of plmDCA (a coevolutionary-based contact predictor)

on real MSAs against those obtained on the same MSAs after a reshuffling that keeps unaltered the position

specific amino acid frequecies and the pairwise Hamming distances between sequences (i.e. Null model II).

As a general comment, the approach looks interesting although I find it hard to understand

whether the conclusions are interesting or not. My major concers follow.

1. The authors evaluate DCA couplings at sequence separation > 4. Such sequence separation

is probably too short to get strong conclusions from the analysis. Residue-residue

contacts are more abundant for short sequence separations and are usually ignored

since they "mask" the more interesting long-range contact.

In fact, contact prediction accuracy is usually assessed on long range contacts

(sequence separation > 24), which provide stronger constraints on the protein 3D structure.

2. It is not clear why the author chose to focus only on 9 Pfam protein families.

Since this work essentually provides a statistical analysis, more robust conclusions

may be obtained on a larger set of protein families. For example, I would suggest

to consider the (quite popular) benchmark set of 150 single chain, single domain

proteins taken from the PSICOV/MetaPSICOV benchmark dataset (obtained from Pfam).

Also, the author should find some way to summarize all the results in a single

plot/table instead of showing a different plot for each protein family. As an alternative, the

authors should at least justify why this cannot be done or why this is not

useful/interesting.

3. In order to make the results more accessible to the scientific community

that works on protein contact prediction it would be useful to

to compare the contact prediction accuracy on the real MSAs against

the accuracy obtained on the shuffled MSAs (e.g. Null model II). In particular, it is not

clear whether plmDCA achieves good or bad prediction accuracy on the benchmark set.

4. The conclusions are a bit misleading. For instance, in Section "Conclusion"

the authors state that "contact prediction is influenced only to a very limited

degree by phylogenetic couplings" but they also add that

"it is also striking that, across several protein families, the phylogeny-caused couplings

in Null models II and III almost reach the DCA-score threshold found before for accurate

contact prediction. This suggests that the suppression of phylogenetic biases in the data

(or their better consideration in model inference), may shift this threshold down and

therefore allow for predicting much more contacts."

First of all, I remark again that 9 protein families are not enough to draw strong conlusions.

Anyway, such concluding remarks do not clarify whether we should care or not

about phylogenetic bias in coevolutionary-based contact prediction.

5. It would be interesting to know (or at least discuss) whether the phylogeny-caused

couplings identified on the shuffled MSAs can be used to filter-out false positive

contact predictions. Maybe I am worng, but at least in principle, it seems to me that

this could be possible by simply performing a Z-test.

6. I would be curious to know how much the null model II shuffles the single

protein sequences. Such analysis coulbe easily done by simply comparing/aligning

a sequence in the shuffled MSA against those in the real MSA in order to detect

the one with the highest sequence similarity. The average sequence similarity can thus

give an idea of how much the shuffled MSA is similar to the real one.

**Have all data underlying the figures and results presented in the manuscript been provided?**

Reviewer #1: Yes

Reviewer #2: None

Reviewer #3: Yes

Reviewer #4: Yes

PLOS authors have the option to publish the peer review history of their article (what does this mean?). If published, this will include your full peer review and any attached files.

Reviewer #1: No

Reviewer #2: **Yes: **Paolo De Los Rios

Reviewer #3: No

Reviewer #4: No
---

## [Decision Letter · Decision Letter 1]

17 Feb 2021

Dear Weigt,

Thank you very much for submitting your manuscript "On the effect of phylogenetic correlations in coevolution-based contact prediction in proteins" for consideration at PLOS Computational Biology. As with all papers reviewed by the journal, your manuscript was reviewed by members of the editorial board and by several independent reviewers. The reviewers appreciated the attention to an important topic. Based on the reviews, we are likely to accept this manuscript for publication, providing that you modify the manuscript according to the review recommendations.

Dear authors, there are still some pending issues raised by one of the reviwer. I encorauge you to take this into consideration and resubmit the manuscript with comments as soon as possble

Sincerely,

Rita Casadio

Guest Editor

PLOS Computational Biology

Arne Elofsson

Deputy Editor

PLOS Computational Biology

[LINK]

Dear authors, there are still some pending issues raised by one of the reviwer. I encorauge you to take this into consideration and resubmit the manuscript with comments as soon as possble

Reviewer's Responses to Questions

**Comments to the Authors:**

Reviewer #1: I thank the authors for the clearer statement of their conclusions and for the improvements of their analyses despite the difficulties of working with bad Internet connections etc. As such the study sheds light on an interesting and not much quantitatively researched aspect of Potts models.

I have two remaining major issues that should be easy to address.

Major point:

1. Starting on line 393, the authors write:

`Suppression of phylogenetic biases in the data (or their better consideration in model inference) ... may allow for predicting much more contacts. The comparison with Null model I, which contains only finite-sample effects at given residue conservations, shows that sufficiently big protein families will be needed to exploit this. For the smallest protein families, represented by our dataset DS3, the finite-sample effects are found to be almost as large as the phylogenetic effects on the DCA results, while the two were well separated in the case of the deeper 400 MSA in datasets DS1 and DS2.'.

Looking at Fig. 3B (showing data for DS3 with small MSAs), at F^APC value corresponding to a PPV of 0.5, the density of null model I predictions in green is about a factor 2 lower than the density of null model II predictions. This means if the phylogenetic biases could be removed, the PPV would increase from TP/(TP+FP) = 0.5 to roughly TP'/(TP'+FP) = TP/(TP+FP/2) = 0.67. This is no small improvement and contradicts the above statement made by the authors.

Obviously, this result is somewhat surprising as the logarithmic y axis hides the magnitude of the effect, and it is cumbersome to read it off the graph. It would therefore be important to add traces to Fig. 5 that show the PPV versus number of predictions for the hypothetical case that phylogenetic effects can be corrected, for DS2 and DS3. The discussion and conclusions will probably have to be adjusted accordingly.

2. Did the authors use sequence weighting? For the comparison of the green and magenta traces in Fig. 3 it is strictly necessary to not use sequence weighting. Otherwise, the weights for null model I would be close to one (since sequences are independent) while those of null model II would be significantly smaller than 1, which would systematically shrink the sizes of the coupling coefficients under null model II and make the distribution incomparable to null model I. Sequence weighting would also render the analysis in point 1 meaningless.

Minor points:

3. Define abbreviation APC, explain in 1-2 sentences what it does and give a proper reference for completeness.

4. Define `normalized frequency' used in Fig. 3.

5. The caption of Figure 4 is missing an explanation what A and B refers to.

6. In figure 4B, what sense does contact and non-contact make for null model II where there are no contacts?

7. Line 363:

`In Fig. 5 we show that the two lead to highly comparable results, with a slight advantage for the standard F^APC after the first few predictions.'

Please explain why. My guess: Residue conservation is positively correlated with both higher variance of the DCA scores and with a higher probability for a contact. Therefore, dividing by the square root of the variance can reduce PPV.

Reviewer #3: I have no further comments/concerns.

Reviewer #4: The revised version of the manuscript addresses all my concerns. I have no further comments at this time.

**Have all data underlying the figures and results presented in the manuscript been provided?**

Reviewer #1: Yes

Reviewer #3: Yes

Reviewer #4: Yes

PLOS authors have the option to publish the peer review history of their article (what does this mean?). If published, this will include your full peer review and any attached files.

Reviewer #1: No

Reviewer #3: No

Reviewer #4: No

Figure Files:

Data Requirements:

Reproducibility:

References:

---

## [Editor Report · Decision Letter 2]

9 Apr 2021

Dear Weigt,

We are pleased to inform you that your manuscript 'On the effect of phylogenetic correlations in coevolution-based contact prediction in proteins' has been provisionally accepted for publication in PLOS Computational Biology.

Best regards,

Arne Elofsson

Deputy Editor

PLOS Computational Biology

Arne Elofsson

Deputy Editor

PLOS Computational Biology

---

## [Editor Report · Acceptance letter]

20 May 2021

PCOMPBIOL-D-20-01261R2 

On the effect of phylogenetic correlations in coevolution-based contact prediction in proteins

Dear Dr Weigt,

I am pleased to inform you that your manuscript has been formally accepted for publication in PLOS Computational Biology. Your manuscript is now with our production department and you will be notified of the publication date in due course.

With kind regards,

Olena Szabo
